# Eggs Are Cost-Efficient in Delivering Several Shortfall Nutrients in the American Diet: A Cost-Analysis in Children and Adults

**DOI:** 10.3390/nu12082406

**Published:** 2020-08-11

**Authors:** Yanni Papanikolaou, Victor L. Fulgoni

**Affiliations:** 1Nutritional Strategies, Nutrition Research & Regulatory Affairs, 59 Marriott Place, Brant, ON N3L 0A3, Canada; 2Nutrition Impact, Nutrition Research, 9725 D Drive North, Battle Creek, MI 49014, USA; vic3rd@aol.com

**Keywords:** NHANES, eggs, cost, nutrients, energy

## Abstract

The purpose of the current study was to examine the cost of eggs in relation to nutrient delivery in children and adults. The present analysis used dietary intake data from the National Health and Nutrition Examination Survey 2013–2016 (egg consumers: 2–18 years-old, *N* = 956; 19+ years-old, *N* = 2424). Inflation adjusted food cost and the cost of nutrients were obtained from the Center for Nutrition Promotion and Policy food cost database. Cost and nutrient profiles for What We Eat in America food categories were compared to whole eggs. Of the 15 main food groups examined, whole eggs ranked third for lowest cost per 100 g (excluding beverages), such that eggs cost 0.35 USD per 100 g, with dairy and grains representing the first and second most cost-efficient foods, at 0.23 USD and 0.27 USD per 100 g, respectively. In children and adults, eggs represented a cost-efficient food for protein delivery, such that eggs provided nearly 2.7% and 3.7% of all protein in the diet, respectively, at a cost of about 0.03 USD per g of protein. Eggs contributed 3.8% and 6.0% of all vitamin A in the diet of children and adults, at a cost of approximately 0.002 USD and 0.003 USD per RAE mcg of vitamin A, respectively. In children 2–18 years-old, nearly 12% of all choline in the diet is delivered from eggs, at a cost of approximately 0.002 USD per mg of choline. Similarly, in adults 19-years-old+, eggs provide nearly 15% of all dietary choline in the diet, at a cost of approximately 0.002 USD per mg of choline. Eggs provide nearly 5% and 9.5% of all vitamin D in the diet of children and adults, at a cost of approximately 0.21 USD and 0.22 USD per mcg of vitamin D, respectively. Overall, eggs ranked as the most cost-efficient food for delivering protein, choline, and vitamin A, second for vitamin E, and third for vitamin D in children. In adults, eggs ranked as the most cost-efficient food for delivering protein and choline, second for vitamin A, and third for vitamin D and vitamin E. In summary, eggs represent an economical food choice for the delivery of protein and several shortfall nutrients (choline, vitamin A, and vitamin D) in the American diet.

## 1. Introduction

The USDA Economic Research Report identifies the majority of US households to be food secure, which is defined as “consistent, dependable access to enough food for active, healthy living”, however, a meaningful percentage of households experience food insecurity at various times throughout the year, often being linked with a lack of economic resources [1]. In 2018, the USDA estimated that 1 in 9 Americans were food insecure, which represents over 37 million Americans, including more than 11 million children [1]. Concurrently, an analysis of food costs during the current Covid-19 pandemic shows that food prices are on the rise. Indeed, the consumer price index (CPI) for all food increased 1.5% between March 2020 and April 2020 and was 3.5% higher compared to April 2019. Furthermore, based on the recent month over month elevation in food costs, USDA has revised the annual forecast and predict that food-at-home prices to increase between 2 and 3% in 2020 [2].

Numerous published studies have demonstrated that food costs can be a predictor of food selection and diet quality, particularly for segments of the population facing economic challenges [3,4,5,6]. In fact, diet costs have been positively linked with scores from the Healthy Eating Index (HEI) [6,7]—a diet quality assessment tool which provides a measure of adherence to the Dietary Guidelines for Americans [8]. Higher diet costs have been associated with higher HEI scores attributed with greater consumption of fruits and vegetables, and lower intake of solid fat, added sugars and alcohol [6]. In contrast, lower food costs have been associated with lower HEI scores supported by lower consumption of fruits, vegetables, whole grains and seafood and greater consumption of solid fat, added sugars, refined grains, and alcohol [7]. A recent cost of food and nutrients analysis using NHANES 2011–2014 identified protein foods as the second most expensive food category, collectively responsible for about 18% and 23% of the daily food costs in children and adults, respectively; however, protein foods were the least expensive sources for choline delivery. Milk and dairy foods accounted for approximately 6% and 12% of daily food costs in children and adults, and were the least expensive sources of calcium and vitamin D in the US diet [9]. Another food economic analysis using the USDA Food and Nutrient Database for Dietary Studies and the Center for Nutrition Policy and Promotion food prices database documented animal-sourced protein, including milk and eggs, as affordable and nutrient-dense food options [4]. Similarly, milk and eggs ranked as the top two economical foods for delivering vitamin D to the American diet [9].

While several studies have examined the costs of nutrients derived from protein foods in the American diet [4,5,6,9,10], there are limited data published on the cost of essential nutrients, including shortfall nutrients, sourced from various protein food sources, with a particular focus on eggs. Eggs have been selected as a focal food in the present research, as the current dietary guidelines have established their nutritional value in several dietary patterns. Therefore, the objectives of the current study were to determine the cost-efficiency of eggs vs. other protein foods in delivering nutrients and energy in the American diet of children and adults.

## 2. Methods

The analysis used data from NHANES—a cross-sectional, nationally-representative survey directed by the National Center for Health Statistics. NHANES samples free-living, non-institutionalized individuals, and is now a continuous study complied by the Centers for Disease Control and Prevention (CDC), where data is released every two years [11]. Ethical protocols, including informed consent from study participants, have been previously obtained, approved and documented by the CDC ethic boards. Twenty-four hour dietary recall data for children 2–18 years-old and adults ≥19 years-old were combined for the present analyses, from two NHANES datasets (i.e., NHANES 2013–2014 and 2015–2016; total *N* for children = *N* = 5669; ≥19 years-old, *N* = 10,112). Data for the nutrients examined are from the U.S. Department of Agriculture (USDA) Food and Nutrient Database for Dietary Studies (FNDDS) database for NHANES [12].

The FNDDS databases determine food and beverage nutrient values in What We Eat in America (WWEIA) [13,14], which is the dietary intake component of NHANES. Twenty-four-hour dietary recalls are collected using the Automated Multiple Pass Method (AMPM) [15]. Although two days of recall are recorded in NHANES, the current analysis focused on 24-h recalls obtained from Day 1, which were collected via an in-person interview. Accuracy, effectiveness, and efficiency of the AMPM method has been extensively reported and previously documented [15].

### 2.1. Egg Consumers and Definition of Eggs

Data that were deemed to be reliable and included completed 24-h recalled dietary data were included in the analyses. Egg consumers (*N* = 956 for children 2–18 years-old; *N* = 2424 for adults ≥19-years-old) were defined as individuals consuming the following eggs and egg-containing foods:(a)All eggs (whole, egg yolk and egg white) with no consideration for eggs in egg-containing dishes, and,(b)All eggs (whole, egg yolk and egg white) + eggs from egg-containing dishes(c)All eggs (whole, egg yolk and egg white) + eggs from egg-containing dishes + eggs from baked goods (i.e., cakes, cookies, brownies, etc.)

Egg intake was determined by FNDDS food codes defined in WWEIA egg categories and egg-containing food categories. Since eggs are also incorporated into mixed dishes, the present analysis separated out eggs in mixed dishes using the Food Patterns Equivalents Databases, which provide the number of ounces of eggs per 100 g of food.

### 2.2. Estimates of Dietary Intake and Percentage of Nutrients from Protein Food Categories

Statistical procedures were completed using SAS software (Version 9.4, SAS Institute, Cary, NC, USA) and SUDAAN 11.0 (RTI International, Research Triangle Park, Durham, NC, USA). The investigation used survey weights to develop nationally representative estimates and adjustment for the complex sample design of NHANEs. Means (±standard errors) for daily intake of nutrients for the daily total diet and from food groups were determined in children and adults, using day 1 dietary intake data. The population ratio method was used to determine the percentage contribution from protein foods. Mean percentage of total dietary intake of nutrients contributed from protein food groups were considered, in addition to mean total energy and nutrient intakes.

### 2.3. Estimates of Food Cost

Food cost and the cost of nutrients were sourced from Center for Nutrition Promotion and Policy (CNPP) 2001–2002 and 2003–2004 cost databases [16,17]. CNPP 2001–2002 and 2003–2004 databases have provided cost per 100 g for foods collected and described in NHANES 2001–2002 and 2003 and 2004, respectively. Food prices for all foods and beverages, as reported in the 2001–2002 and 2003–2004 cycles of NHANES, were adjusted for inflation using monthly Consumer Price Index from the Bureau of Labor Statistics [2]. Additionally, the water category was comprised of four categories (7702 ‘Tap water’, 7704 ‘Bottled water’, 7802 ‘Flavored or carbonated water’ and 7804 ‘Enhanced or fortified water’). Bottled water was assigned a cost of $0.25 per 100 g/L, based on a recent sale price of private label bottled water. Tap water was assigned a cost of 1/300th the cost of bottled water based upon online estimates of the relative costs of tap and bottled water. For flavored, carbonated, enhanced or fortified water, the costs are based on the same methodology used for other foods, using the inflation adjusted cost database. The average costs as $ per 100 g for water within WWEIA food categories used in the analysis were as follows: tap water (0.000083); bottled water (0.025); flavored or carbonated water (0.060); enhanced or fortified water (0.152).

## 3. Results

### 3.1. Cost of Food Groups and Protein Foods in the American Diet

Table 1 and Table 2 present the cost of main food groups and protein food subgroups, respectively, per 100 g of food. Of USDA 15 main food groups, whole eggs rank #3 for cost-effectiveness per 100 g (excluding beverages).

Table 3 and Table 4 present the daily cost of main food groups and protein food subgroups, respectively, in children. Milk and dairy, protein foods and mixed dishes comprise approximately 54% of daily food costs in children, while protein foods (excluding milk and dairy and mixed dishes) alone represent 19% of all food costs in children 2–18-years-old. Similarly, Table 5 and Table 6 present the daily cost of main food groups and protein food subgroups, respectively, in adults. Milk and dairy, protein foods and mixed dishes comprise approximately 52% of daily food costs. Protein foods (excluding dairy and mixed dishes) alone represent 24% of all food costs in American adults. Eggs comprise about 1.2% and 1.5% of daily food costs in children and adults respectively (Table 4 and Table 6). Costs are also provided for all USDA main food groups and protein food subgroups.

### 3.2. Cost of Nutrients from USDA Protein Food Groups

Table 7, Table 8, Table 9, Table 10, Table 11, Table 12, Table 13 and Table 14 depict percent daily nutrient contribution from eggs in children and adults. Furthermore, the tables provide cost per unit of nutrient for eggs and other protein-containing food groups (i.e., milk and dairy and protein foods). Eggs provide nearly 2.7% of all protein in the diet of children, at a cost of approximately $0.03 per g of protein (Table 7). In adults, eggs contribute nearly 3.7% of all protein in the diet, at a cost of approximately $0.03 per g of protein (Table 8). Eggs provide nearly 12% and 15% of all dietary choline in the diet of children and adults, respectively, at a cost of $0.002 per mg of choline. Approximately 5% and 9.5% of all vitamin D in the diet of children and adults are sourced from eggs, at a cost of approximately $0.21 and $0.22 per mcg of vitamin D, respectively. Similarly, eggs provide nearly 4% and 6% of all vitamin A in the diet of children and adults, at a cost of approximately $0.002 and $0.003 per RAE mcg of vitamin A. Eggs contributed nearly 2.5% and 3.5% of all vitamin E in the diet of children and adults, respectively, at a cost of approximately $0.31 per mg of vitamin E.

While the current analysis examined several sources of nutrients and the cost of nutrients, smaller percentages of certain nutrients were provided by eggs. For example, eggs provided nearly 1.1% and 1.5% of all potassium in the diet of children and adults, respectively, at a cost of approximately $0.002 per mg of potassium. Eggs contributed about 1.6% and 2.6% of all dietary iron in the diet of children and adults, respectively, at a cost of approximately $0.26 per mg of iron. Likewise, eggs contributed nearly 1% of all magnesium in the diet of children and adults, at a cost of approximately $0.03 per mg of magnesium.

## 4. Discussion

During challenging economic times, many Americans can benefit from nutrition research that focuses on cost-nutrient analyses to determine cost-effective sources for foods and nutrients. To our knowledge, this is the first study to demonstrate the cost-efficiency of eggs in providing nutrients in the American diet. As cost has been identified as a key barrier in preventing many Americans from consuming recommended nutrients from healthy eating patterns [3,4,5,6], eggs offer a cost-effective food choice in the deliverance of nutrients to the diet of children and adults. The current data show that eggs cost approximately $0.35 per 100 g, and eggs are a cost-effective food for the delivery of dietary protein, choline, vitamin D, vitamin A and vitamin E. In children, eggs ranked as the most cost-effective food for delivering protein, choline, and vitamin A, second for vitamin E, and third for vitamin D. Similarly, in adults, eggs ranked as the most cost-effective food for delivering protein and choline, second for vitamin A, and third for vitamin D and vitamin E.

Current dietary guidelines advocate for a variety of protein foods, of which include eggs, particularly when consumed with limited sodium, solid fat, and added sugar [18]. Previous literature has documented the nutrient-rich aspect of eggs, such that one egg (50 g serving) contributes numerous bioactives and essential nutrients [19,20]. Further, eggs are an important dietary source of choline, an essential nutrient that is under-consumed by the American population [21]. A 50 g egg contributes 147 mg of dietary choline, and has been documented as a leading food source for choline in the American diet [22]. The metabolic, physiological and structural importance of choline has been well documented by previous reviews and the National Academy of Medicine (previously known as the Institute of Medicine) [21,23,24]. With choline involved in neurotransmitter synthesis, lipid transport, metabolic regulation, detoxification, cell signaling, and as an essential building block for cell membranes, NHANES data support the fact that the majority of Americans, including children and adolescents, are not meeting established recommendations for choline intake, thus, creating a potential public health concern regarding choline intake [21]. A recent review highlighted a lack of data availability on usual intake of choline throughout the lifespan, with particular emphasis on children, and the need for future research initiatives to fill such nutrient research gaps [24]. According to the National Institutes of Health, the majority of the US population consume less than the Adequate Intake for choline, while it is estimated that 90–95% of pregnant women have choline intakes below the recommendations [25]. NHANES data has also reported that African American males had lower dietary choline intakes relative to other race/ethnicities, while choline intakes did not differ in female counterparts [26]. Our current data show that eggs contribute nearly 12% and 15% of all dietary choline in the diet of children and adults, respectively. Further, our analysis shows that eggs offer the most cost-effective approach in choline delivery—eggs provide choline at a cost of $0.002 per mg of choline, with the second most cost-effective food being milk at $0.008 per mg of choline.

Eggs are also a dietary source for bioavailable lutein and zeaxanthin, with a 50 g egg contributing approximately 250 µg lutein + zeaxanathin [22]. Lutein and zeaxanthin are carotenoids with accumulating evidence linking to eye health and a reduced risk for vision-related diseases [27,28,29,30]. While the current study did not examine the contribution of lutein + zeaxanthin, our previous research in children and adolescents using NHANES data show that a dietary pattern that includes eggs is associated with significantly greater daily lutein + zeaxanthin intake, in addition to higher intake of protein, polyunsaturated, monounsaturated and total fat, α-linolenic acid, docosahexaenoic acid (DHA), choline, vitamin D, potassium, phosphorus, and selenium [20].

The 2015–2020 Dietary Guidelines for Americans (DGA) [18] and the guiding scientific report issued by the 2015 Dietary Guidelines Scientific Advisory Committee (DGAC) [31] have identified several nutrients currently under-consumed in Americans ≥2 years of age, relative to the Estimated Average Requirement or Adequate Intake levels set by the former Institute of Medicine (i.e., National Academy of Medicine)—the shortfall nutrients identified include vitamin A, vitamin D, vitamin E, vitamin C, folate, calcium, magnesium, fiber, potassium in all Americans, and iron for adolescent and premenopausal females. In the current analysis, eggs have been shown to provide meaningful amounts of three DGA/DGAC shortfall nutrients. Additionally, our data show the cost-effectiveness of consuming eggs in delivering vitamin A, D and E, on a per unit nutrient basis.

As has been documented previously in observational studies similar to the present analysis [19,20], our study has several limitations and strengths characteristic of epidemiological research. The current dietary pattern was assessed using dietary recall data from NHANES, which contributes a distinctive advantage to researchers, through access to a large cross-sectional database that combines sophisticated, in-person assessments, with validated biochemical, clinical and anthropometric examinations. In addition, a large, nationally representative database to assess the cost of nutrients from foods is a distinctive strength. Limitations of the current analysis include memory recall bias, including the under- or over-reporting of foods and beverages; however, measures have been incorporated to minimize errors in data collection. An additional limitation in the present work may be linked to food prices that do not reflect geographic variations in food price and overall dietary costs. Further strengths and limitations have been previously published and discussed [32,33,34,35].

## 5. Conclusions

The current data show that eggs cost approximately $0.35 per 100 g, representing a cost-efficient food for delivery of dietary protein, choline, vitamin D, vitamin A and vitamin E. In children, eggs ranked as the most cost-efficient food for delivering protein, choline, and vitamin A, second for vitamin E, and third for vitamin D. Similarly, in adults, eggs are ranked as the most economical food for delivering protein and choline, second for vitamin A, and third for vitamin D and vitamin E. In summary, eggs represent a cost-efficient food choice for the delivery of protein and several shortfall nutrients in the American diet of children and adults.

## Figures and Tables

**Table 1 nutrients-12-02406-t001:** Mean cost ($) and cost-efficient ranking of USDA main food groups per 100 g compared to a whole egg.

Main Food Groups	Cost ($ Per 100 g)	Rank (Including Beverages)	Rank (Excluding Beverages)
Water	0.01	1	-
Beverages, non-alcoholic	0.07	2	-
Beverages, alcoholic	0.21	3	-
Milk and Dairy	0.23	4	1
Grains	0.27	5	2
Whole Eggs	0.35	6	3
Vegetables	0.37	7	4
Infant Formula and Baby Food	0.37	7	4
Sugars	0.38	8	5
Fruit	0.39	9	6
Condiments and Sauces	0.47	10	7
Mixed Dishes	0.48	11	8
Fats and Oils	0.48	11	8
Snacks and Sweets	0.64	12	9
Protein Foods	0.99	13	10
Other	1.32	14	11

**Table 2 nutrients-12-02406-t002:** Mean cost of USDA dairy and protein food subgroups compared to eggs.

Protein Food Category	Cost ($ Per 100 g)	Rank
Milk	0.13	1
Flavored Milk	0.16	2
Dairy Drinks and Substitutes	0.24	3
Yogurt	0.37	4
Eggs	0.39	5
Plant-based Protein Foods	0.58	6
Poultry	0.82	7
Cheese	0.95	8
Cured Meats/Poultry	0.96	9
Protein and Nutritional Powders	1.33	10
Meats	1.54	11
Seafood	2.06	12

Of USDA protein food subgroups examined, whole eggs rank #3 for cost-effectiveness per 100 g (excluding beverages).

**Table 3 nutrients-12-02406-t003:** Estimated percentage of children, age 2–18 years, who consume specified food groups on a given day, and the estimated mean daily cost ($) of each food group expressed in dollars, and as a percentage of the total cost of food.

Main Food	Cost			
Groups	Mean	SE	% Daily	SE
All	4.75	0.06	100.00	0.000
Milk and Dairy	0.51	0.01	10.6	0.29
Protein Foods	0.90	0.03	18.8	0.60
Mixed Dishes	1.18	0.02	24.8	0.47
Grains	0.30	0.01	6.3	0.21
Snacks and Sweets	0.68	0.02	14.2	0.30
Fruit	0.34	0.01	7.1	0.28
Vegetables	0.23	0.01	4.8	0.17
Beverages, non-alcoholic	0.39	0.01	8.3	0.23
Alcoholic Beverages	0.02	0.01	0.3	0.14
Water	0.08	0.00	1.6	0.08
Fats and Oils	0.04	0.00	0.9	0.07
Condiments and Sauces	0.06	0.00	1.2	0.06
Sugars	0.02	0.00	0.5	0.05
Infant Formula and Baby Food	0.01	0.00	0.3	0.06
Other	0.01	0.00	0.2	0.08

NHANES 2013–2016; children 2–18 years-old; Day 1 Intake; ‘% Daily’ refers to the percent daily contribution of the nutrient from the food group on a given day.

**Table 4 nutrients-12-02406-t004:** Estimated percentage of children, age 2–18 years, who consume dairy and other protein food subgroups on a given day, and the estimated mean daily cost ($) of each food subgroup expressed in dollars and as a percentage of the total cost of food.

Main Food	Cost			
Subgroups	Mean	SE	% Daily	SE
Milk	0.22	0.008	4.7	0.2
Flavored Milk	0.09	0.006	1.8	0.1
Dairy Drinks and Substitutes	0.03	0.003	0.6	0.1
Cheese	0.12	0.006	2.4	0.1
Yogurt	0.05	0.005	1.0	0.1
Meats	0.24	0.016	5.0	0.3
Poultry	0.25	0.017	5.3	0.3
Seafood	0.12	0.017	2.5	0.3
Eggs	0.06	0.003	1.2	0.1
Cured Meats/Poultry	0.17	0.010	3.5	0.2
Plant-based Protein Foods	0.06	0.005	1.3	0.1

NHANES 2013–2016; children 2–18 years-old; Day 1 Intake; ‘% Daily’ refers to the percent daily contribution of the nutrient from the food group on a given day.

**Table 5 nutrients-12-02406-t005:** Estimated percentage of adults, age ≥ 19 years, who consume specified food groups on a given day, and the estimated mean daily cost ($) of each food group expressed in dollars, and as a percentage of the total cost of food.

Main Food Groups	Cost			
	Mean	SE	% Daily	SE
All	6.49	0.064	100.0	0.0
Milk and Dairy	0.40	0.010	6.1	0.1
Protein Foods	1.54	0.050	23.7	0.6
Mixed Dishes	1.41	0.026	21.8	0.4
Grains	0.29	0.007	4.5	0.1
Snacks and Sweets	0.55	0.013	8.4	0.2
Fruit	0.34	0.014	5.2	0.2
Vegetables	0.48	0.011	7.4	0.2
Beverages, Non-alcoholic	0.66	0.015	10.1	0.2
Alcoholic Beverages	0.41	0.024	6.4	0.4
Water	0.14	0.005	2.2	0.1
Fats and Oils	0.10	0.004	1.6	0.1
Condiments and Sauces	0.12	0.006	1.8	0.1
Sugars	0.03	0.001	0.4	0.02
Infant Formula and Baby Food	0.0004	0.0003	0.006	0.004
Other	0.02	0.003	0.3	0.05

NHANES 2013–2016; adults ≥19 years-old; Day 1 Intake; ‘% Daily’ refers to the percent daily contribution of the nutrient from the food group on a given day.

**Table 6 nutrients-12-02406-t006:** Percent daily cost ($) of eggs vs. other protein food subgroups in adults.

Main Food	Cost			
Subgroups	Mean	SE	% Daily	SE
Milk	0.13	0.005	1.9	0.1
Flavored Milk	0.01	0.002	0.2	0.0
Dairy Drinks and Substitutes	0.04	0.004	0.5	0.1
Cheese	0.16	0.007	2.4	0.1
Yogurt	0.06	0.004	1.0	0.1
Meats	0.39	0.017	6.0	0.3
Poultry	0.31	0.015	4.8	0.2
Seafood	0.36	0.036	5.6	0.5
Eggs	0.10	0.004	1.5	0.1
Cured Meats/Poultry	0.23	0.009	3.5	0.1
Plant-based Protein Foods	0.15	0.008	2.3	0.1

NHANES 2013–2016; adults ≥19 years-old; Day 1 Intake; ‘% Daily’ refers to the percent daily contribution of the nutrient from the food group on a given day.

**Table 7 nutrients-12-02406-t007:** Percent daily protein (g) and cost ($) of protein from eggs in children compared to USDA protein food subgroups.

Main Food	Protein					
Subgroups	Mean	SE	% Daily	SE	Cost/Unit	SE
Milk	6.10	0.22	9.0	0.31	0.037	0.001
Flavored Milk	1.73	0.12	2.5	0.18	0.050	0.001
Dairy Drinks and Substitutes	0.32	0.04	0.5	0.06	0.092	0.006
Cheese	2.45	0.14	3.6	0.21	0.048	0.0005
Yogurt	0.71	0.07	1.0	0.11	0.070	0.002
Meats	4.23	0.25	6.2	0.35	0.056	0.001
Poultry	7.15	0.43	10.5	0.59	0.035	0.001
Seafood	1.19	0.17	1.7	0.25	0.101	0.004
Eggs	1.83	0.09	2.7	0.14	0.031	0.0003
Cured Meats/Poultry	3.19	0.18	4.7	0.26	0.052	0.001
Plant-based Protein Foods	1.49	0.10	2.2	0.15	0.041	0.002

Data are for children 2–18 years-old; NHANES 2013–2016, Day 1 data. Cost/Unit = mean cost of one unit of nutrient (i.e., protein) from foods in the category; ‘% Daily’ refers to the percent daily contribution of the nutrient from the food group on a given day.

**Table 8 nutrients-12-02406-t008:** Percent daily protein (g) and cost ($) of protein from eggs in adults compared to USDA protein food subgroups.

Main Food	Protein					
Subgroups	Mean	SE	% Daily	SE	Cost/Unit	SE
Milk	3.15	0.12	3.8	0.14	0.040	0.001
Flavored Milk	0.27	0.03	0.3	0.04	0.054	0.002
Dairy Drinks and Substitutes	0.28	0.03	0.3	0.04	0.128	0.010
Cheese	3.20	0.14	3.9	0.16	0.049	0.001
Yogurt	1.14	0.08	1.4	0.09	0.057	0.001
Meats	6.94	0.31	8.4	0.36	0.056	0.001
Poultry	9.01	0.40	10.9	0.44	0.035	0.0005
Seafood	3.56	0.33	4.3	0.38	0.103	0.004
Eggs	3.09	0.11	3.7	0.13	0.032	0.0003
Cured Meats/Poultry	4.35	0.18	5.2	0.21	0.053	0.001
Plant-based Protein Foods	3.09	0.13	3.7	0.16	0.048	0.001

Data are for adults ≥19 years-old; NHANES 2013–2016, Day 1 data. Cost/Unit = mean cost of one unit of nutrient (i.e., protein) from foods in the category; ‘% Daily’ refers to the percent daily contribution of the nutrient from the food group on a given day.

**Table 9 nutrients-12-02406-t009:** Percent daily choline (mg) and cost ($) of choline from eggs in children compared to USDA protein food subgroups.

Main Food	Choline					
Subgroups	Mean	SE	% Daily	SE	Cost/Unit	SE
Milk	29.79	1.08	11.9	0.39	0.008	0.0001
Flavored Milk	8.66	0.60	3.5	0.24	0.010	0.0001
Dairy Drinks and Substitutes	2.19	0.27	0.9	0.11	0.013	0.001
Cheese	2.69	0.15	1.1	0.06	0.043	0.001
Yogurt	2.01	0.19	0.8	0.07	0.025	0.0004
Meats	13.33	0.82	5.3	0.33	0.018	0.0005
Poultry	19.75	1.30	7.9	0.49	0.013	0.0002
Seafood	4.32	0.63	1.7	0.25	0.028	0.001
Eggs	29.69	1.71	11.8	0.61	0.002	0.00005
Cured Meats/Poultry	10.87	0.62	4.3	0.24	0.015	0.0004
Plant-based Protein Foods	4.90	0.32	2.0	0.13	0.013	0.001

Data are for children 2–18 years-old; NHANES 2013–2016, Day 1 data. Cost/Unit = mean cost of one unit of nutrient (i.e., choline) from foods in the category; ‘% Daily’ refers to the percent daily contribution of the nutrient from the food group on a given day.

**Table 10 nutrients-12-02406-t010:** Percent daily choline (mg) and cost ($) of choline from eggs in adults compared to USDA protein food subgroups.

Main Food	Choline					
Subgroups	Mean	SE	% Daily	SE	Cost/Unit	SE
Milk	15.23	0.60	4.5	0.16	0.008	0.0002
Flavored Milk	1.41	0.16	0.4	0.05	0.010	0.0004
Dairy Drinks and Substitutes	1.90	0.23	0.6	0.07	0.019	0.002
Cheese	3.35	0.13	1.0	0.04	0.047	0.001
Yogurt	2.82	0.18	0.8	0.05	0.023	0.0004
Meats	23.01	1.08	6.8	0.33	0.017	0.0004
Poultry	24.57	1.13	7.3	0.35	0.013	0.0001
Seafood	13.91	1.25	4.1	0.36	0.026	0.001
Eggs	50.04	1.74	14.8	0.44	0.002	0.00004
Cured Meats/Poultry	14.56	0.68	4.3	0.19	0.016	0.0004
Plant-based Protein Foods	10.37	0.44	3.1	0.12	0.014	0.001

Data are for adults ≥19 years-old; NHANES 2013–2016, Day 1 data. Cost/Unit = mean cost of one unit of nutrient (i.e., choline) from foods in the category; ‘% Daily’ refers to the percent daily contribution of the nutrient from the food group on a given day.

**Table 11 nutrients-12-02406-t011:** Percentage daily vitamin D (mcg) and cost ($) of vitamin D from eggs in children, compared to USDA protein food subgroups.

Main Food	Vitamin D					
Subgroups	Mean	SE	% Daily	SE	Cost/Unit	SE
Milk	2.27	0.08	42.3	0.8	0.098	0.001
Flavored Milk	0.63	0.05	11.8	0.8	0.137	0.002
Dairy Drinks and Substitutes	0.08	0.01	1.6	0.2	0.349	0.032
Cheese	0.28	0.02	5.3	0.3	0.411	0.023
Yogurt	0.11	0.01	2.0	0.2	0.463	0.028
Meats	0.04	0.00	0.7	0.1	6.278	0.515
Poultry	0.05	0.00	1.0	0.1	4.723	0.172
Seafood	0.22	0.04	4.0	0.8	0.554	0.068
Eggs	0.27	0.02	5.1	0.3	0.207	0.005
Cured Meats/Poultry	0.11	0.01	2.1	0.1	1.459	0.052
Plant-based Protein Foods	0.0004	0.0002	0.007	0.004	160.26	85.91

Data are for children 2–18 years-old; NHANES 2013–2016, Day 1 data. Cost/Unit = mean cost of one unit of nutrient (i.e., vitamin D) from foods in the category; vitamin D = vitamin D2 + D3; ‘% Daily’ refers to the percent daily contribution of the nutrient from the food group on a given day.

**Table 12 nutrients-12-02406-t012:** Percent daily vitamin D (mcg) and cost ($) of vitamin D from eggs in adults, compared to USDA protein food subgroups.

Main Food	Vitamin D					
Subgroups	Mean	SE	% Daily	SE	Cost/Unit	SE
Milk	1.18	0.05	25.0	0.9	0.106	0.002
Flavored Milk	0.09	0.01	2.0	0.2	0.159	0.006
Dairy Drinks and Substitutes	0.11	0.01	2.3	0.3	0.329	0.018
Cheese	0.26	0.01	5.6	0.3	0.597	0.033
Yogurt	0.13	0.01	2.7	0.2	0.510	0.019
Meats	0.07	0.01	1.6	0.1	5.267	0.355
Poultry	0.04	0.00	0.9	0.1	7.304	0.278
Seafood	0.80	0.09	17.0	1.6	0.457	0.053
Eggs	0.44	0.01	9.5	0.3	0.220	0.004
Cured Meats/Poultry	0.14	0.01	3.1	0.2	1.581	0.059
Plant-based Protein Foods	0.001	0.0002	0.02	0.005	142.47	34.77

Data are for adults ≥19 years-old; NHANES 2013–2016, Day 1 data. Cost/Unit = mean cost of one unit of nutrient (i.e., vitamin D) from foods in the category; vitamin D = vitamin D2 + D3; ‘% Daily’ refers to the percent daily contribution of the nutrient from the food group on a given day.

**Table 13 nutrients-12-02406-t013:** Percent daily vitamin A (RAE mcg) and cost ($) of vitamin A from eggs in children, compared to USDA protein food subgroups.

Main Food	Vitamin A					
Subgroups	Mean	SE	% Daily	SE	Cost/Unit	SE
Milk	100.17	3.73	16.9	0.5	0.002	0.00004
Flavored Milk	31.17	2.21	5.3	0.4	0.003	0.0001
Dairy Drinks and Substitutes	9.26	1.09	1.6	0.2	0.003	0.0001
Cheese	29.23	1.70	4.9	0.3	0.004	0.0001
Yogurt	3.34	0.61	0.6	0.1	0.015	0.002
Meats	1.11	0.72	0.2	0.1	0.214	0.136
Poultry	4.11	0.28	0.7	0.1	0.062	0.002
Seafood	1.34	0.23	0.2	0.0	0.089	0.007
Eggs	22.71	1.25	3.8	0.2	0.002	0.0001
Cured Meats/Poultry	1.45	0.27	0.2	0.0	0.115	0.021
Plant-based Protein Foods	0.61	0.26	0.1	0.0	0.101	0.042

Data are for children 2–18 years-old; NHANES 2013–2016, Day 1 data. Cost/Unit = mean cost of one unit of nutrient (i.e., vitamin A) from foods in the category; RAE = retinol activity equivalents; ‘% Daily’ refers to the percent daily contribution of the nutrient from the food group on a given day.

**Table 14 nutrients-12-02406-t014:** Percent daily vitamin A (RAE mcg) and cost ($) of vitamin A from eggs in adults, compared to USDA protein food subgroups.

Main Food	Vitamin A					
Subgroups	Mean	SE	% Daily	SE	Cost/Unit	SE
Milk	51.69	2.02	8.2	0.3	0.002	0.00005
Flavored Milk	4.30	0.55	0.7	0.1	0.003	0.0002
Dairy Drinks and Substitutes	9.23	0.97	1.5	0.2	0.004	0.0001
Cheese	35.50	1.44	5.6	0.2	0.004	0.0001
Yogurt	5.40	0.52	0.9	0.1	0.012	0.001
Meats	5.66	1.97	0.9	0.3	0.069	0.024
Poultry	5.08	0.39	0.8	0.1	0.061	0.003
Seafood	4.53	0.46	0.7	0.1	0.080	0.004
Eggs	37.91	1.25	6.0	0.2	0.003	0.0001
Cured Meats/Poultry	5.00	1.84	0.8	0.3	0.046	0.016
Plant-based Protein Foods	0.65	0.15	0.1	0.0	0.224	0.048

Data are for adults ≥19-years-old; NHANES 2013–2016, Day 1 data. Cost/Unit = mean cost of one unit of nutrient (i.e., vitamin A) from foods in the category; RAE = retinol activity equivalents; ‘% Daily’ refers to the percent daily contribution of the nutrient from the food group on a given day.

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
