# Peer review of "Eggs Are Cost-Efficient in Delivering Several Shortfall Nutrients in the American Diet: A Cost-Analysis in Children and Adults"

_nutrients, 2020, doi:10.3390/nu12082406_

Round 1
Reviewer 1 Report
Manuscript # nutrients-855078 “Eggs Represent a Cost-Effective Approach in Delivering Several Shortfall Nutrients in the American Diet of Children and Adults”
The manuscript describes results from a cross-section study of two waves of the NHANES (13-14, 15-16), with a combined sample of 5,669 children and 10,112 adults. The primary objective was to determine the cost-effectiveness of eggs vs. other protein foods in delivering nutrients and energy in the American diet. While the research question is potentially important, it seems that the justification of the particular approach is not sufficiently clear. In addition, the term cost-effectiveness has specific meaning in economic evaluation of health care interventions (to assist resource allocation). In the context of eggs being one of the lower cost protein sources, it would be more appropriate to consider a cost-minimization analysis (to delivery a given level of protein, what would be the costs of using different sources)? Please refer to the reference for some background information: Weinstein MC, Siegel JE, Gold MR, Kamlet MS, Russell LB, for the Panel on Cost-Effectiveness in Health and Medicine. Recommendations of the
panel on cost-effectiveness in health and medicine. JAMA. 1996;276:1253-58.
A number of main concerns related to the methodology should be addressed. 1) The main issue is that given the poor population nutritional status (i.e. obesity across all subgroups of the US population), the cross-sectional analysis of the reported consumption pattern from 2013-2016 would be misleading in terms of finding. 2) Given the recent publication by Hess et al., the manuscript does not seem to offer additional information, as the fact that egg is a cost-efficient protein source was generally accepted. Perhaps an examination of the association between egg/protein consumption and unhealthy foods consumption would be more fruitful; 3) As important as proteins are, the premise of the study that food costs can be a predictor of diet quality, while a reasonable one, do not capture the complexity of the issue: low cost unhealthy foods may crowd out high cost of healthy foods, especially for individuals who come from disadvantaged socioeconomic background; 4) The tables included a lot of information but not enough evaluation of what do them mean for the consumers, researchers, and policymakers.
Introduction:
Page 2, line 48, please clarify the direction of the association.
Line 52, “to” should be changed to “with.”
Line 68, please provide a better justification of the current study. The benefits of eggs were never brought into question, given the extent literature. The main nutritional issue is the consumption of foods with questionable nutritional value, therefore it is unclear how potentially increasing egg consumption, while beneficial, could help deal with the issue.
Line 75, it is unclear why two waves of data are needed, given the more than sufficient size of sample in either wave.
Line 88, please clarify the basis of the classification and to what extent it helps or hinders the purpose of this analysis.
Page 3, line 97, The three ways of quantifying egg consumptions (a, b, c) highlights the main issue with the current approach: while a and arguably b are generally healthy use of eggs, eggs from baked foods are clearly not (because eggs are used with large quantities of sugar in these foods). This is a societal problem though.
Line 116, Was the food component of the CPI used?
Table 1, liquids (water, beverages, and milk) should be excluded from this table altogether. In addition, cost alone is not a good representation of the value of the food categories. It is somewhat shocking that beverages, sugars, condiments/sauces, snacks and sweets are even included, as they arguably are not foods in the sense that people can live without.
Table 2, Liquids should be excluded from the per 100g calculation.
Table 3, the table in its current format is not very informative. Perhaps it should be ranked by % consumers in descending order? Liquids should be separated because the unit used in calculating costs. Are costs in $? Please label the unit of the mean column.
The large number (14) of tables appears unjustified. The authors should consider summarizing the main findings in the text, or consolidate the tables, or move some tables to appendices.
Page 5, Table 5, there seems to be a lack of critical evaluation of the food categories/groups used in the table. For example, the results that 77.2% of consumers reported eating snacks and sweets and 92.4% beverages should sound have sounded an alarm regarding the dire status of the American diet. This is not an issue of costs, as these questionable items are often inexpensive.
Page 8, line 216. Please add a citation regarding cost has a key barrier, as it is clear from the authors’ analysis that protein sources are not necessarily more expensive (such as eggs). Thus the poor quality of the American diet must have other major contributing factors at play.
Line 224, this is an excellent point. The fact of the matter is not so much as whether people eat eggs, but what they are eating eggs with (sodium, fat, and added sugar). Thus a more nuanced approach is necessary to make the case for increased consumption of eggs.
Author Response
Thank you for taking the time to provide insightful feedback. We have addressed all of your questions and comments in bold font below. Please let us know if you have additional comments and/or questions.
Reviewer Feedback:
The manuscript describes results from a cross-section study of two waves of the NHANES (13-14, 15-16), with a combined sample of 5,669 children and 10,112 adults. The primary objective was to determine the cost-effectiveness of eggs vs. other protein foods in delivering nutrients and energy in the American diet. While the research question is potentially important, it seems that the justification of the particular approach is not sufficiently clear. In addition, the term cost-effectiveness has specific meaning in economic evaluation of health care interventions (to assist resource allocation). In the context of eggs being one of the lower cost protein sources, it would be more appropriate to consider a cost-minimization analysis (to delivery a given level of protein, what would be the costs of using different sources)? Please refer to the reference for some background information: Weinstein MC, Siegel JE, Gold MR, Kamlet MS, Russell LB, for the Panel on Cost-Effectiveness in Health and Medicine. Recommendations of the panel on cost-effectiveness in health and medicine. JAMA. 1996;276:1253-58.
Authors’ Response: Thank you for your thoughtful comments. We agree with your idea of examining a cost-minimization research approach and have initiated discussions to complete such research as a follow-up to the current manuscript. We are interested in the concept of examining protein delivery costs from various animal and plant protein sources. We think it’s a novel idea. Also, for the current manuscript, we have revised the term “cost-effective” to “cost-analysis” and appreciate your sharing the JAMA background material. Finally, we have also revised the justification in the introduction to read as follows:
“While several studies have examined the costs of nutrients derived from protein foods in the American diet [4,5,6,9,10], there are limited data published on the cost of essential nutrients, including shortfall nutrients, sourced from various protein food sources, with particular focus on eggs. Eggs have been selected as a focal food in the present research, as the current dietary guidelines have established their nutritional value in several dietary patterns. Therefore, the objectives of the current study was to determine the cost-efficiency of eggs vs. other protein foods in delivering nutrients and energy in the American diet of children and adults.”
The title has also been revised to “Eggs are Cost-Efficient in Delivering Several Shortfall Nutrients in the American Diet: A Cost-Analysis in Children and Adults”
A number of main concerns related to the methodology should be addressed. 1) The main issue is that given the poor population nutritional status (i.e. obesity across all subgroups of the US population), the cross-sectional analysis of the reported consumption pattern from 2013-2016 would be misleading in terms of finding. 2) Given the recent publication by Hess et al., the manuscript does not seem to offer additional information, as the fact that egg is a cost-efficient protein source was generally accepted. Perhaps an examination of the association between egg/protein consumption and unhealthy foods consumption would be more fruitful; 3) As important as proteins are, the premise of the study that food costs can be a predictor of diet quality, while a reasonable one, do not capture the complexity of the issue: low cost unhealthy foods may crowd out high cost of healthy foods, especially for individuals who come from disadvantaged socioeconomic background; 4) The tables included a lot of information but not enough evaluation of what do them mean for the consumers, researchers, and policymakers.
Authors’ Response: Thanks for providing these 4 comments. We will address them individually below:
- We are not promoting egg consumption as a method to help lower obesity rates in the US population. Our research aims to demonstrate that eggs are a cost-efficient food choice for delivering several nutrients underconsumed by the US population. Please let us know if we have missed anything from your comment. Thank you.
- We agree that the Hess et al. paper was an important contribution to the literature. A key point raised by Hess et al. using NHANES 2011-2014 is that “…this work reinforce the importance of consuming a variety of nutrient-rich foods for cost-effective, sustainable eating patterns. In our current research, we aimed to further expand the area of focus into egg consumption and cost-efficiency in delivering shortfall nutrients. The current analysis also used a more recent NHANES database to provide an updated cost-analysis.
- We agree in that more research is required. Our future projects are evaluating how low cost foods are associated with diet quality and potential nutritional consequences that may arise. For example, disadvantaged socioeconomic backgrounds likely lack fish and seafood protein, which may have nutritional implications (i.e.,. omega-3 intake)—these are areas of interest for our group and research that can be examined using different databases, including NHANES. We have previously published a paper sparking an interest in this area—the previous publication is: Comparison of Inadequate Nutrient Intakes in Non-Hispanic Blacks vs. Non-Hispanic Whites: An Analysis of NHANES 2007-2010 in U.S. Children and Adults, J Health Care Poor Underserved, 2015. 26(3): 726-736.
- We agree that we have provided a substantial amount of information in the tables. We prefer to keep the tables as is, since previously reviewed work has requested having more vs. less information, so that readers can have substantial information available for other relevant projects. Additionally, the first 4 tables have been revised and reformatted to address less decimal places and edited titles. If you approve of the formatting, we can make all final changes to remaining tables in the final manuscript.
Please let us know if you have further questions or comments. Thank you.
Introduction:
Page 2, line 48, please clarify the direction of the association.
Authors’ Response: A revision to this section has been made as recommended to include the positive direction of the association. Thank you.
Line 52, “to” should be changed to “with.”
Authors’ Response: This correction has been completed. Thank you.
Line 68, please provide a better justification of the current study. The benefits of eggs were never brought into question, given the extent literature. The main nutritional issue is the consumption of foods with questionable nutritional value, therefore it is unclear how potentially increasing egg consumption, while beneficial, could help deal with the issue.
Authors’ Response: We appreciate your thorough review of our work. We have revised the justification for the present analysis in the last paragraph of the introduction. We are not emphasizing increasing egg consumption, but rather that eggs can be part of a healthy dietary pattern. Our goal was to publish this data to help alleviate any misconceptions around eggs and health, and also demonstrate that eggs can be a cost-efficient approach to delivery of several shortfall nutrients. Further, eggs are contributors of several shortfall nutrients, in addition, to being large contributors of choline and lutein+zeaxanthin. Please let us know if you have further comments or questions.
Line 75, it is unclear why two waves of data are needed, given the more than sufficient size of sample in either wave.
Authors’ Response: We have included two waves of data to ensure a large sample size. This process has been consistently followed in previous research from both our group and the research of other groups. Please let us know if you have further comments. Thank you.
Line 88, please clarify the basis of the classification and to what extent it helps or hinders the purpose of this analysis.
Authors’ Response: This statement was included to provide further context on the USDA classification. We now see that this information is likely not important for the reader and as such, we have deleted from the manuscript. Thank you for pointing this out to us.
Page 3, line 97, The three ways of quantifying egg consumptions (a, b, c) highlights the main issue with the current approach: while a and arguably b are generally healthy use of eggs, eggs from baked foods are clearly not (because eggs are used with large quantities of sugar in these foods). This is a societal problem though.
Authors’ Response: Yes, we agree with your observation. While eggs from bakery foods are higher in sugar, saturated fat and sodium, we wanted to complete a cost-analysis that considered all egg-containing foods in the diet.
Line 116, Was the food component of the CPI used?
Authors’ Response: Yes, the food component of the CPI was used. Let us know if you have further questions. Thank you.
Table 1, liquids (water, beverages, and milk) should be excluded from this table altogether. In addition, cost alone is not a good representation of the value of the food categories. It is somewhat shocking that beverages, sugars, condiments/sauces, snacks and sweets are even included, as they arguably are not foods in the sense that people can live without.
Authors’ Response: Based on your feedback and feedback from the other reviewer, we have made revisions to Table 1. Based on the other reviewer feedback, additional information was requested for water, which has been also added into the methods section.
Table 2, Liquids should be excluded from the per 100g calculation.
Authors’ Response: We prefer to leave the liquids in Table 2 to allow for comparisons to dairy foods and dairy substitutes. Please let us know if you want to discuss further. Thank you.
Table 3, the table in its current format is not very informative. Perhaps it should be ranked by % consumers in descending order? Liquids should be separated because the unit used in calculating costs. Are costs in $? Please label the unit of the mean column.
Authors’ Response: We have added “$” to cost. Thank you for catching this error. We have also adjusted the table to meet your recommendations for descending order. If you are in approval of this format, we can adjust all tables to follow this format in the final manuscript. We encourage keeping liquids with the other foods. If you disagree, please let us know. Thank you.
The large number (14) of tables appears unjustified. The authors should consider summarizing the main findings in the text, or consolidate the tables, or move some tables to appendices.
Authors’ Response: We included 14 tables as previous reviewers have suggested more tables and less text. We are willing to move some tables to appendices if it is your preference and the journal’s recommendation. We are open to both formats. Thank you.
Page 5, Table 5, there seems to be a lack of critical evaluation of the food categories/groups used in the table. For example, the results that 77.2% of consumers reported eating snacks and sweets and 92.4% beverages should sound have sounded an alarm regarding the dire status of the American diet. This is not an issue of costs, as these questionable items are often inexpensive.
Authors’ Response: We agree and have seen from several of our other NHANES analyses that there are alarming attributes with the current American dietary pattern. For Table 5, while 92.4% reported consuming beverages, we don’t describe the type of beverages consumed. Table 5 is meant to display the cost of the main food groups as classified by USDA as a general overview. If you feel that this table can be removed, we would consider adding the information into the text. Please let us know if you have further comments. Thank you.
Page 8, line 216. Please add a citation regarding cost has a key barrier, as it is clear from the authors’ analysis that protein sources are not necessarily more expensive (such as eggs). Thus the poor quality of the American diet must have other major contributing factors at play.
Authors’ Response: Thank you for this comment. Several citations have been added to support this statement involving ‘cost as a key barrier’.
Line 224, this is an excellent point. The fact of the matter is not so much as whether people eat eggs, but what they are eating eggs with (sodium, fat, and added sugar). Thus a more nuanced approach is necessary to make the case for increased consumption of eggs.
Authors’ Response: Thank you for this acknowledgement and we appreciate your comments and helpful recommendations to improve the manuscript.
Reviewer 2 Report
Substantive comments:
Line 108—Please explain in detail how the percentages were calculated. The population ratio method should have been used. Was it?
Lines 111-117 and lines 153-154—Something is wrong with the cost of food estimates. They should not differ between adults and children. The cost of 1 egg is the cost of 1 egg regardless of how it is consumed or who eats it.
Line 121 and following—“Cost-effectiveness” implies some sort of positive effect. Here the effect is providing 100 grams of food, and that’s not very meaningful. Only the cost per unit of the various nutrients should be provided. Tables 1 and 2 should be omitted.
Table 3—
- The title is unclear. It should be something like “Estimated percentage of children, age 2-18 years, who consume specified food groups on a given day and the estimated mean daily cost of each food group expressed in dollars and as a percent of the total cost of food.”
- Add a footnote defining “Consumer %,” that is, percent consuming on a given day.
- Add the unit of measure for Cost Mean, that is, $.
- Add a footnote explaining “% Daily.”
- Similar changes should be made in all tables.
Tables 3 and 5, water row--It is suspicious that the cost of water is $0.08 and $0.14 per day for children and adults, respectively--higher than many other food groups. Please check how the price of water was determined, and explain in the Methods section.
Discussion section is much too long. Please focus it on the topic of this paper. This is not the place to share everything known about eggs. Consider returning to the theme of food security and the relatively low cost of protein from eggs.
Editorial comments:
All tables have too many decimal places. The data are not that precise.
Line 48—Omit “USDA’s.” The National Cancer Institute has been an equal partner in the development and evaluation of the HEI since the 2005 version.
Line 49—Please be more specific. Change “authoritative dietary guidance” to the “the Dietary Guidelines for Americans.”
Line 50—Change “Elevated” to “Higher.”
Line 52—Change “reduced” to “lower.” “Reduced” implies a change over time.
Line 60-62—The cited paper (reference #8) does not include the analysis described here.
Line 69—Change “Experimental Section” to “Methods.” This study was not an experiment.
Line 81—Change “represents” to “is.”
Lines 81-84—Change “The collection procedure for WWEIA involves use of the Automated Multiple Pass Method (AMPM), representing a dietary collection tool that provides a valid, evidence-based approach for gathering data for national dietary surveys [15]” to “Twenty-four-hour dietary recalls are collected using the Automated Multiple Pass Method (AMPM) [15].”
Lines 100-102—Change “where USDA provides serving amounts of eggs for all foods with the assumption that a fixed amount of energy and nutrients from eggs were incorporated into calculations” to “which provide the number of ounces of eggs per 100 grams of food.” The rest of the sentence doesn’t make sense. Please explain what the authors did more clearly.
Line 123—The title of Table 1 does not match its content.
Line 128-129—Complaining about how USDA classifies foods into groups serves no purpose here. Most readers will already know that dairy products have their own food group because of their calcium content. Please omit this phrase.
Lines 151 and 153—Omit “USDA.” These are not the USDA Protein Food groups/subgroups.
Line 152—Define or explain “meaningful” or omit it.
Line 165—Change “were sourced from” to “were provided by.”
Tables 7-15—For each nutrient, specify what the unit is; for example, grams for protein.
Author Response
Thank you for providing insightful feedback during the review process. We have addressed each question/comment individually below and welcome any further feedback you may have. The authors' responses are in bolded font for ease of viewing.
Reviewer Feedback
Line 108—Please explain in detail how the percentages were calculated. The population ratio method should have been used. Was it?
Authors’ Response: Yes, the population ratio method was used to determine the percentage contribution from protein food and the methods have been changed to include this clarification. We have also added additional information for water calculations in the methods section. Thank you for bringing this to our attention.
Lines 111-117 and lines 153-154—Something is wrong with the cost of food estimates. They should not differ between adults and children. The cost of 1 egg is the cost of 1 egg regardless of how it is consumed or who eats it.
Authors’ Response: This is a good question! Children and adults will have different cost of food estimates since different total daily food budgets are allocated to children vs. adults, as are different total amounts of food consumed. In Table 1, “Cost of USDA dairy and food subgroups compared to a whole egg”, the analysis looks at cost of whole eggs regardless of how eggs are consumed or who eats the eggs. Please let us know if you have further questions or comments.
Line 121 and following—“Cost-effectiveness” implies some sort of positive effect. Here the effect is providing 100 grams of food, and that’s not very meaningful. Only the cost per unit of the various nutrients should be provided. Tables 1 and 2 should be omitted.
Table 3—
- The title is unclear. It should be something like “Estimated percentage of children, age 2-18 years, who consume specified food groups on a given day and the estimated mean daily cost of each food group expressed in dollars and as a percent of the total cost of food.”
- Add a footnote defining “Consumer %,” that is, percent consuming on a given day.
- Add the unit of measure for Cost Mean, that is, $.
- Add a footnote explaining “% Daily.”
- Similar changes should be made in all tables.
Authors’ Response:
- We agree with your recommendation and agree that the title suggested is clearer. The title has been revised to “Estimated percentage of children, age 2-18 years, who consume specified food groups on a given day and the estimated mean daily cost of each food group expressed in dollars and as a percent of the total cost of food”. Thank you.
- The footnote defining “consumer %” has been made throughout the document. Thank you.
- The unit of measure has been added as per your recommendation in all relevant tables. Thank you.
- A footnote has been added as suggested. Thank you.
- Changes have been incorporated in all tables. Thank you.
Tables 3 and 5, water row--It is suspicious that the cost of water is $0.08 and $0.14 per day for children and adults, respectively--higher than many other food groups. Please check how the price of water was determined, and explain in the Methods section.
Authors’ Response: The water category was comprised of four categories ( 7702 'Tap water', 7704 'Bottled water', 7802 'Flavored or carbonated water' and 7804 'Enhanced or fortified water'). Bottled water was assigned a cost of $0.25 per 100 grams/liter based on a recent sale price of private label bottled water. Tap water was assigned a cost of 1/300th the cost of bottled water based upon online estimates of the relative costs of tap and bottled water. For flavored, carbonated, enhanced or fortified water the costs are based on the same methodology used for other foods using the inflation adjusted cost database.
For additional background, the average costs as $ per 100 grams for water within What We Eat in American (WWEIA) food categories used in the analysis were as follows:
Tap water 0.000083
Bottled water 0.025
Flavored or carbonated water 0.060
Enhanced or fortified water 0.152
These details have been now added to the methods section. Please let us know if you have further questions. Thank you.
Discussion section is much too long. Please focus it on the topic of this paper. This is not the place to share everything known about eggs. Consider returning to the theme of food security and the relatively low cost of protein from eggs.
Authors’ Response: We agree with your comment regarding the discussion. We have reviewed and shortened the discussion (i.e., removal of approximately 200 words) as per your recommendation. We encourage leaving in the section on choline, particularly since the draft report released by the 2020 Dietary Guidelines Scientific Advisory Committee emphasizes the importance of choline in the diet. Please let us know if you have additional questions or comments.
Editorial comments:
All tables have too many decimal places. The data are not that precise.
Authors’ Response: We are revising the tables and formatting so that the final manuscript version will have less decimal places as recommended. Please see Tables 1-4 for all changes (including titles of tables) and let us know if this will be acceptable and we can make final changes to remaining tables. Thank you.
Line 48—Omit “USDA’s.” The National Cancer Institute has been an equal partner in the development and evaluation of the HEI since the 2005 version.
Authors’ Response: “USDA’ has been removed and agree with your comment. Thank you.
Line 49—Please be more specific. Change “authoritative dietary guidance” to the “the Dietary Guidelines for Americans.”
Authors’ Response: We have revised to delete “authoritative dietary guidance” to “the Dietary Guidelines for Americans”
Line 50—Change “Elevated” to “Higher.”
Authors’ Response: We have changed “elevated” to “higher”. Thank you.
Line 52—Change “reduced” to “lower.” “Reduced” implies a change over time.
Authors’ Response: We agree and have made the edit as suggested. Thank you.
Line 60-62—The cited paper (reference #8) does not include the analysis described here.
Authors’ Response: This has been corrected to reflect reference 4 which is the Drewnowski 2010 AJCN study. Thank you for catching this.
Line 69—Change “Experimental Section” to “Methods.” This study was not an experiment.
Authors’ Response: The change has been made as recommended. Thank you for this observation.
Line 81—Change “represents” to “is.”
Authors’ Response: The change from “represents” to “is” has been completed.
Lines 81-84—Change “The collection procedure for WWEIA involves use of the Automated Multiple Pass Method (AMPM), representing a dietary collection tool that provides a valid, evidence-based approach for gathering data for national dietary surveys [15]” to “Twenty-four-hour dietary recalls are collected using the Automated Multiple Pass Method (AMPM) [15].”
Authors’ Response: Thank you for this suggestion. We agree and have revised accordingly.
Lines 100-102—Change “where USDA provides serving amounts of eggs for all foods with the assumption that a fixed amount of energy and nutrients from eggs were incorporated into calculations” to “which provide the number of ounces of eggs per 100 grams of food.” The rest of the sentence doesn’t make sense. Please explain what the authors did more clearly.
Authors’ Response: We have revised the sentence as per your recommendation and have revised the sentence to clarify for the reader.
Line 123—The title of Table 1 does not match its content.
Authors’ Response: We have revised the title to read “Mean cost ($) and cost-efficient ranking of USDA food subgroups per 100 g compared to a whole egg.”
Line 128-129—Complaining about how USDA classifies foods into groups serves no purpose here. Most readers will already know that dairy products have their own food group because of their calcium content. Please omit this phrase.
Authors’ Response: Our intention was never to complain about how USDA classifies food. We only intended to help orient the reader. However, we don’t want to be misperceived as such, thus, we are removing the statement as per your recommendation. Thank you for your observation and comments.
Lines 151 and 153—Omit “USDA.” These are not the USDA Protein Food groups/subgroups.
Authors’ Response: We have omitted “USDA” as per your recommendation.
Line 152—Define or explain “meaningful” or omit it.
Authors’ Response: The word “meaningful” has been deleted. Thank you.
Line 165—Change “were sourced from” to “were provided by.”
Authors’ Response: This has been changed as per your recommendation. Thank you.
Tables 7-15—For each nutrient, specify what the unit is; for example, grams for protein
Authors’ Response: Thank you for catching this missing information. We have added the unit description for each nutrient in all relevant tables.
Round 2
Reviewer 1 Report
Thank you for addressing my comments. I recognize that you have a very specific goal for the manuscript, which may preclude you from including additional analyses that I was suggesting.